# Wild jackdaws can selectively adjust their social associations while preserving valuable long-term relationships

Michael Kings [1] ✉, Josh J. Arbon [1,2] ✉, Guillam E. McIvor [1], Martin Whitaker[3], Andrew N. Radford [2], Jürgen Lerner [4,5] & Alex Thornton [1] ✉

Influential theories of the evolution of cognition and cooperation posit that tracking information about others allows individuals to adjust their social associations strategically, re-shaping social networks to favour connections between compatible partners. Crucially, to our knowledge, this has yet to be tested experimentally in natural populations, where the need to maintain long-term, fitness-enhancing relationships may limit social plasticity. Using a social-network-manipulation experiment, we show that wild jackdaws (*Corvus monedula*) learned to favour social associations with compatible group members (individuals that provided greater returns from social foraging interactions), but resultant change in network structure was constrained by the preservation of valuable pre-existing relationships. Our findings provide insights into the cognitive basis of social plasticity and the interplay between individual decision-making and social-network structure.

The advantages of adjusting social associations flexibly to maximize rewards are thought to promote the evolution of cognition[1,2] and cooperation[3,4]. In dynamic social environments, individuals may encounter a range of potential social partners and the payoffs of associating with a particular partner can vary over time and across contexts[5]. Thus, the ability to recognize individuals, remember the outcomes of past social interactions and learn the value of different partners could allow individuals to maximize gains from social interactions by retaining associations that prove valuable and discarding those that do not[6,7]. As a by-product of these individual partner-choice decisions, compatible individuals should become increasingly likely to share common social partners, eventually generating self-sustaining clusters within the group's social network and favouring the persistence of cooperation within groups[6–9]. Understanding the ability of animals to optimize social interactions ("social competence"[5]) and the resultant plasticity of social networks is therefore a critical aim of cognitive, behavioural and evolutionary research[6,7,10]. However, empirical progress has been limited by difficulties in quantifying dynamic social adjustments and their network-level consequences in wild populations[11,12].

In natural populations, short-term partner-choice decisions may be affected not only by up-to-date appraisals of group members' current value[6] but also by the need to maintain long-term relationships. Indeed stable, long-lasting cooperative relationships, commonly between kin or individuals that share a common interest, such as monogamous mating partners, are a common feature of many animal societies[13–15]. Quantifying the relative influence of flexible, short-term associations and long-term relationships (akin to 'selective' and 'structural' assortment[16]) on current partner-choice outcomes, and ultimately social-network plasticity, is necessary to understand how the nature of individual decision-making dictates the structure of social groups[17]. To date, relevant experimental work has largely been limited to economic games played amongst unfamiliar people[8,9] or to interactions between laboratory animals[18,19], neither of which fully reflect patterns of association in freely interacting social networks (but see[20]). By contrast, manipulating access to foraging sites[21,22], removing

[1]Centre for Ecology and Conservation, University of Exeter, Penryn Campus, Treliever Road, Penryn TR10 9FE, UK. [2]School of Biological Sciences, University of Bristol, 24 Tyndall Avenue, Bristol BS8 1TQ, UK. [3]technologywithin, Chevron Business Park, Limekiln Lane, Holbury, Southampton, SO45 2QL, UK. [4]Department of Computer and Information Science, University of Konstanz, 78457 Konstanz, Germany. [5]HumTec Institute, RWTH Aachen University, 52062 Aachen, Germany. ✉e-mail: mkings1024@gmail.com; josh.arbon@bristol.ac.uk; alex.thornton@exeter.ac.uk

individuals from populations[23] or observing how individuals respond to demographic changes[24] can provide insights into how group members alter their social preferences in response to perturbations in group connectedness or composition. However, these approaches lack the fine-scale experimental control required to test hypotheses concerning the flexibility of partner-choice decision-making. Moreover, to the best of our knowledge, no study of non-human animals has explicitly tested the role of learning in re-wiring social networks. We used a social-network-manipulation experiment to investigate (a) whether animals in a natural population learn to adjust their social associations selectively to maximize rewards and (b) whether such decisions affect social structure by leading to the formation of clusters of compatible individuals within the network.

We used an automated social coordination task (cf.[25]) to manipulate the value of social foraging associations in a population of wild jackdaws (*Corvus monedula*) fitted with radio frequency identification (RFID) tags (see Study species and site in "Methods" for details). We then observed how ties within the network of jackdaw social foraging associations were re-configured in response to task pay-offs. Jackdaws are highly social, colony-breeding corvids that form long-term, strictly monogamous pair bonds[26]. Offspring retain close, prolonged associations with their parents post-fledging, and siblings from the same brood commonly associate together in creches and at foraging sites[27]. Jackdaws cooperate with other colony members to deter predators[28] and engage in social foraging with both kin (i.e., siblings, parents and their offspring) and non-kin[27,29], often forming large flocks that exhibit fission–fusion dynamics. We classified dyads within long-term relationships (mated pairs; parent–offspring; siblings) as "affiliated" and dyads outside these relationships as "unaffiliated". All birds could engage freely with pairs of automated feeders that responded to combinations of individuals via their RFID-tag codes (Fig. 1a). Individuals were pseudo-randomly assigned to one of two incompatible treatment classes (A and B). Dyads from the same treatment class could cooperate for mutual benefit (*sensu*[30]) by simultaneously occupying feeding positions on the apparatus to gain access to a high-quality food reward (see Supplementary Note 1: Food preference test); hereafter, a "successful" event. Associating with a member of the other treatment class incurred a cost, in the form of triggering a two-minute period of feeder inactivity (see Dual-feeder task in "Methods" and Fig. 1b for details). Treatment-class assignments of known affiliates were constrained such that an equal number of affiliated dyads were rewarded or disincentivized for associating (see Treatment class assignment in "Methods").

By making task pay-offs dependent on participant pairings, we assess whether jackdaws learn to exploit the novel 'biological market'[4] conditions we experimentally impose, without interfering with social group composition. We find that wild jackdaws learn to associate preferentially with compatible individuals (those that provide greater foraging returns), leading to an increase in the overall frequency of associations between compatible social partners over time. As a by-product of individual learning and partner-choice, we observe clustering of compatible individuals within the social foraging network. However, the magnitude of this effect is small, likely due to the preservation of pre-existing long-term relationships. These results provide critical field evidence that learning to adjust social associations is beneficial whilst highlighting trade-offs with the need to maintain valuable long-term relationships.

## Results

### Relational event models

We used Relational Event Models (REMs[31]) to analyze changes in individual performance, patterns of association, within-dyad coordination of behaviour and clustering within the social network according to treatment class. REMs are a form of time-to-event analysis uniquely suited to the study of behavioural dynamics[32]. They provide estimates (hazard ratios, here presented as Incidence Rate Ratios; IRR) of how many times more or less likely a particular type of event was to occur relative to an estimated baseline rate of interaction for the period in question (calculated here from permuted datasets, see Statistical analysis in "Methods" for details). For instance, an IRR value of 1.2 represents a 20% increase in incidence rate relative to the baseline, whilst a value of 0.8 represents a 20% decrease. Confidence intervals

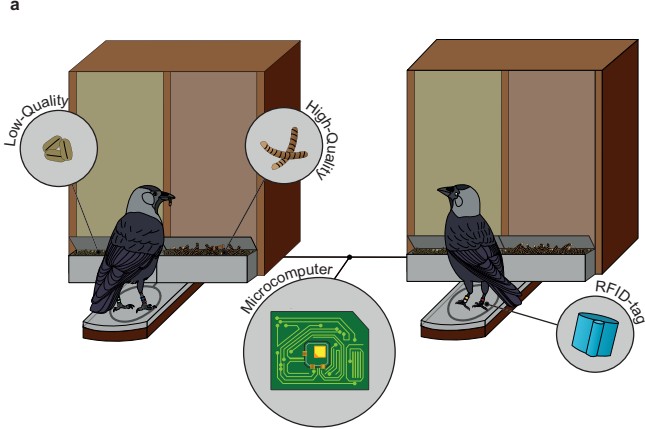

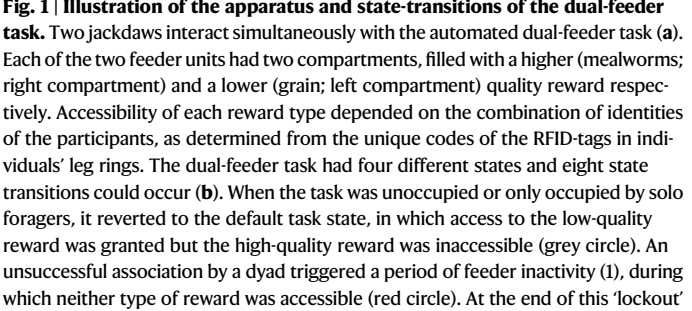

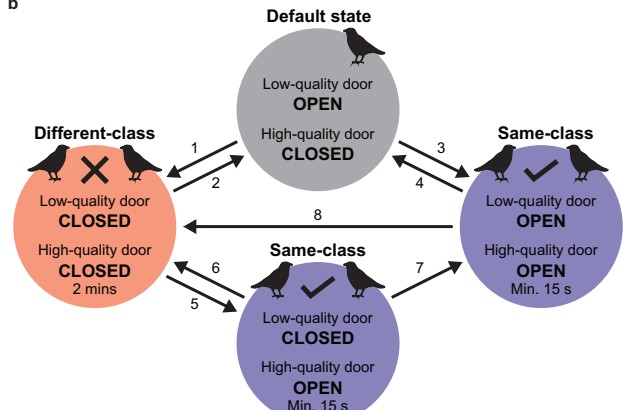

**Fig. 1 | Illustration of the apparatus and state-transitions of the dual-feeder task.** Two jackdaws interact simultaneously with the automated dual-feeder task (**a**). Each of the two feeder units had two compartments, filled with a higher (mealworms; right compartment) and a lower (grain; left compartment) quality reward respectively. Accessibility of each reward type depended on the combination of identities of the participants, as determined from the unique codes of the RFID-tags in individuals' leg rings. The dual-feeder task had four different states and eight state transitions could occur (**b**). When the task was unoccupied or only occupied by solo foragers, it reverted to the default task state, in which access to the low-quality reward was granted but the high-quality reward was inaccessible (grey circle). An unsuccessful association by a dyad triggered a period of feeder inactivity (1), during which neither type of reward was accessible (red circle). At the end of this 'lockout' period, the task returned to its default state (2). A successful dyadic interaction that occurred during a period when a lockout was not in effect resulted in access to both the high- and low-quality rewards (3). The end of a successful event during a non-lockout period resulted in a return to the default task state (4). A successful dyadic interaction that occurred during a lockout period triggered override of the lockout condition and access to the high-quality reward only (5). A successful dyadic interaction that ended before the completion of a lockout period caused the task to revert to the lockout state (6). If a lockout period ended during an ongoing successful event, then the dyad would gain access to both food rewards (7). Finally, an unsuccessful association event that immediately followed a successful event resulted in a direct transition to a lockout period, bypassing the default task state (8).

around the IRR that do not overlap a value of one are taken to indicate statistically significant effects[33].

## Individual performance: learning to adjust social ties

Patterns of visitation reflected the dynamic fission–fusion structure of jackdaw societies, with birds visiting the task, often in groups of variable size, throughout its period of operation (see Supplementary Note 2). We recorded a total of 3117 associations involving 139 individuals across 751 different dyads over the course of four months (see Supplementary Note 3 and Supplementary Fig. 1 for details). Overall, the percentage of successful events (55.4%) significantly exceeded the range of expected values had treatment class combinations of participants occurred at random (95% Confidence Interval [CI] = (49.4, 52.4)), as calculated from 10,000 permuted datasets. Discrimination of task partners based on compatibility was linked to individual success. Individuals that avoided repeatedly associating with incompatible participants ($IRR_{per-event} = 0.997$, CI = (0.996, 0.998)) whilst maximizing repeated associations with compatible participants ($IRR_{per-event} = 1.0015$, CI = (1.0011, 1.0019)) were most likely to be observed engaging in successful events in the future (Supplementary Table 1). Individual success relied on the adjustment of social ties (Fig. 2), as better task performance was exhibited by individuals that rapidly severed ties with incompatible task partners ($IRR_{per-partner} = 0.960$, CI = (0.950, 0.971)) and increased the number of compatible partners with which they associated repeatedly ($IRR_{per-partner} = 1.015$, CI = (1.003, 1.027), Supplementary Table 2). Same-class dyads were, partly by chance (see Fig. 5b), less likely to be observed than different-class dyads at the outset of the experiment (IRR = 0.846, CI = (0.779, 0.917)), but their frequency increased over time, becoming approximately 20% more likely to be observed than different-class associations with each 1000 events (Fig. 3; median per-1000-events IRR ($IRR_{1000}$) = 1.21, CI = (1.15, 1.27), Supplementary Table 3). These analyses show that jackdaws learnt to discriminate between compatible and incompatible participants, associating more frequently with individuals that belonged to the same treatment class and reducing their associations with members of the other class over time. An interesting open question is whether such changes in associations during social foraging carry over into different behavioural contexts. Such reputational effects may be expected if individuals are highly consistent[34] but may be less likely if individuals adjust their behaviour flexibly by tracking context-specific payoffs. Examining the relationship between social flexibility and cross-contextual reputation will be an important focus for future research.

## Patterns of association: effects of pre-existing relationships

Our study also reveals that changing patterns of association through learning were affected by pre-existing, long-term relationships. Overall, prior experience of interacting with a given partner was linked to a greater probability of interacting with the same partner in the future. This effect was stronger for affiliated dyads (1–5 prior associations: IRR = 3.32, CI = (2.62, 4.26); six or more prior associations: IRR = 26.1, CI = (21.1, 32.3), Supplementary Table 4) than for unaffiliated dyads (1–5 prior associations: IRR = 1.35, CI = (1.24, 1.46); six or more prior associations: IRR = 1.12, CI = (1.02, 1.23), Supplementary Table 5). Critically, patterns of continued interaction only reflected the expected response to the experimental treatment when they involved unaffiliated dyads (n = 2469 events involving 139 individuals across 733 dyads) and not when they featured affiliates (n = 648 events involving 24 individuals across 18 dyads). After associating at least once at a feeder together, unaffiliated dyads from the same treatment class (who were rewarded for associating) were approximately 20% more likely to associate in the future as compared to unaffiliated dyads from different treatment classes with equivalent prior task experience (1–5 prior associations: IRR = 1.19, CI = (1.06, 1.35); six or more prior associations: IRR = 1.34, CI = (1.17, 1.53)). Inspection of predicted association rates derived from this REM emphasizes the treatment-dependent change in non-affiliate association as dyads gained task experience (Fig. 4a). By contrast, associations among affiliates were unaffected by

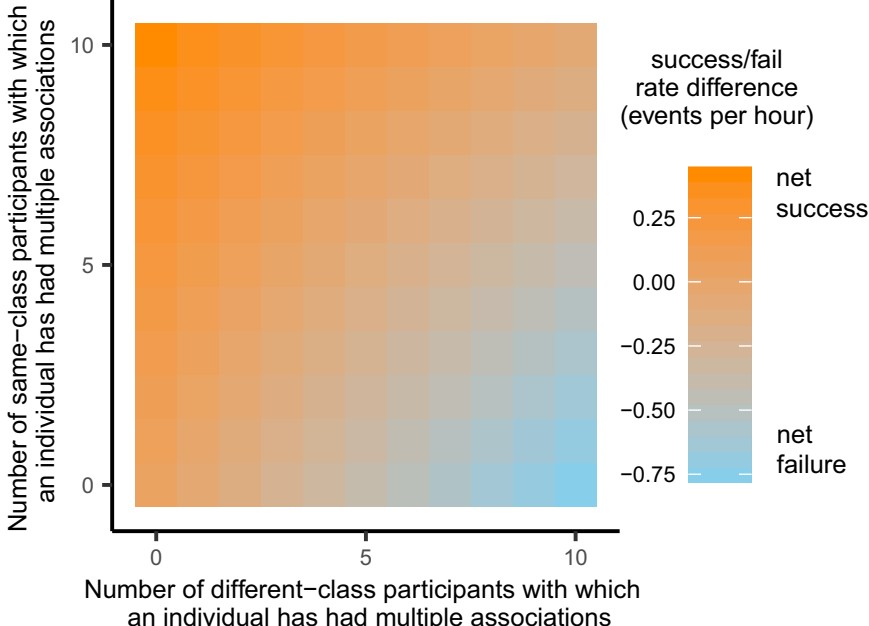

**Fig. 2 | The relationship between individual performance and discrimination of task partners.** Heatmap of the predicted likelihood of an individual engaging in a successful event in the future given the number of different-class and same-class task partners with which it has associated on multiple occasions. Likelihood of future success is given as the change in number of expected successful association events per hour relative to an estimate of the baseline rate of successful events per individual per hour (approximately 2.3 events per hour) calculated across all periods of the experiment (see Supplementary Methods: Visualization of REM output). Likelihood of future success was linked to partner-discrimination: the best-performing individuals avoided repeated associations with different-class participants, whilst increasing the number of same-class participants with which they associated multiple times.

whether dyad members were from the same- or different-treatment classes (1–5 prior associations: IRR = 1.27, CI = (0.85, 1.89); six or more prior associations: IRR = 0.89, CI = (0.64, 1.24)), though model predictions suggest that unsuccessful associations were more likely to have been observed amongst the most experienced affiliate dyads (Fig. 4b). This influence of established relationships on the response to the experimental treatment was also reflected in the trends in incidence rate for different dyad types as the experiment progressed (Fig. 5).

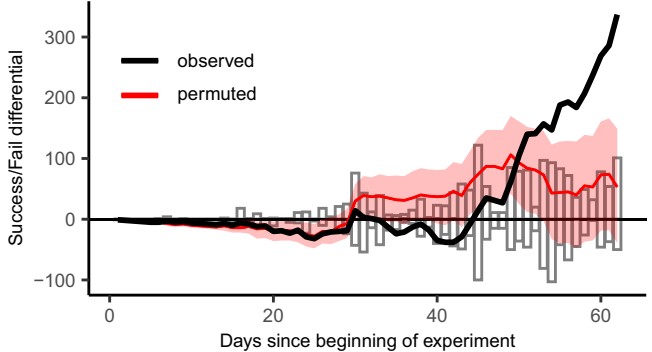

**Fig. 3 | Increase in the ratio of successful to unsuccessful association events as the experiment progressed.** Histograms display daily counts of the number of successful (positive bars) and unsuccessful (negative bars) association events. The difference between these two counts yielded a daily success/fail differential and the cumulative sum of these differences is displayed (black line) to show how the balance of successful versus unsuccessful events changed over time. The median expected cumulative success/fail differential, as calculated from permuted datasets, is also displayed (red line) along with its 95% confidence interval (red shading; for details see 'Permutation tests' in "Methods": Statistical analysis).

## Coordination of dyadic associations

From a cognitive perspective, the ability to recognize the need for a social partner and coordinate actions accordingly may facilitate effective collaboration. To test this, we examined whether compatible partners learned to synchronize the timing of their arrivals at feeders (i.e., reduce the latency between the arrival of each partner) and to spend more time at feeders together. Neither arrival latencies nor association durations differed between same- and different-class partners (latency: $\text{IRR}_{\text{per-second}} = 0.997$, CI = (0.990, 1.004), Supplementary Table 6; duration: $\text{IRR}_{\text{per-second}} = 0.997$, CI = (0.994, 1.001), Supplementary Table 7). Thus, although jackdaws clearly learned to adjust their social associations, we found no evidence that that they learned to coordinate their activities with compatible participants.

## Clustering by treatment class

We did find evidence that the adjustment of individual social associations had knock-on effects for broader social network structure. Convergence of the social neighbourhoods of compatible participants occurred as the experiment progressed: though dyads for which the two members shared fewer common same-class associates were more likely to be observed at the beginning of the experiment ($\text{IRR}_{\text{per-associate}} = 0.972$, CI = (0.954, 0.990), Supplementary Table 8), over time same-class dyads with a greater number of common same-class associates became more likely to be observed than same-class dyads with few common same-class associates ($\text{IRR}_{\text{per-associate, per-1000-events}} = 1.015$, CI = (1.005, 1.024); Fig. 6). Overall, same-class dyads tended to have few common same-class associates (mean = 2.63, 95% CI = (2.28, 2.97)) but the likelihood of observing a same-class dyad with an additional common same-class associate increased by 1.5% per 1000 association events observed. Importantly, no such trend was found when the class composition of common associates was disregarded ($\text{IRR}_{\text{per-associate, per-1000-events}} = 0.998$, CI = (0.995, 1.001)). This indicates that the slight increase in the similarity of individuals' social neighbourhoods was restricted to groupings of compatible (i.e., same-class) participants. However, due to the small magnitude of the effect,

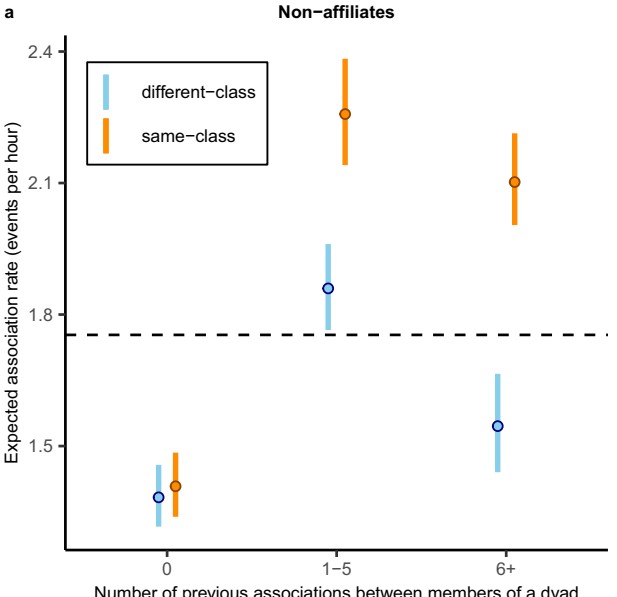

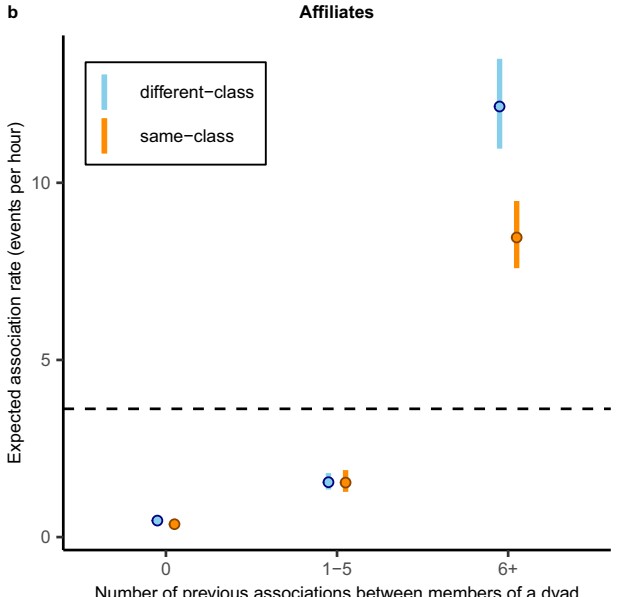

**Fig. 4 | Assortment by treatment class as dyads gained experience of the task.** Change in likelihood of observing same-class (orange) or different-class (blue) dyads as prior task experience increased, for dyads composed of **a** non-affiliates (*n* = 733 dyads, 2469 events) and **b** affiliates (*n* = 18 dyads, 648 events). Likelihood of observation is given in number of expected events per hour, as calculated by multiplying predicted incidence rate ratios by the estimated baseline rate of events per dyad per hour (dashed line) for non-affiliate dyads (1.75 events per hour) or affiliate dyads (3.62 events per hour). Median predicted values and 95% prediction intervals are displayed (for further details see Supplementary Methods: Visualization of REM output).

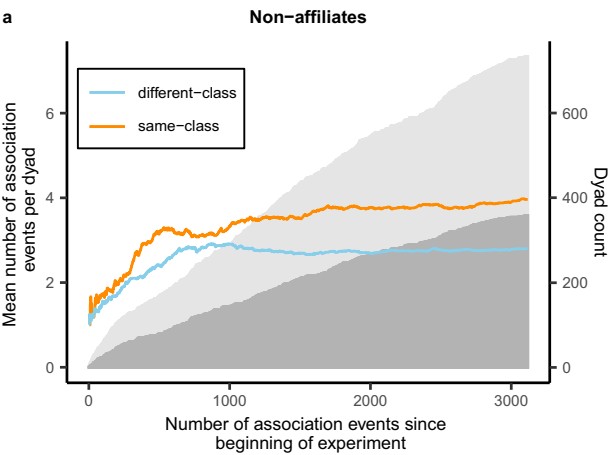

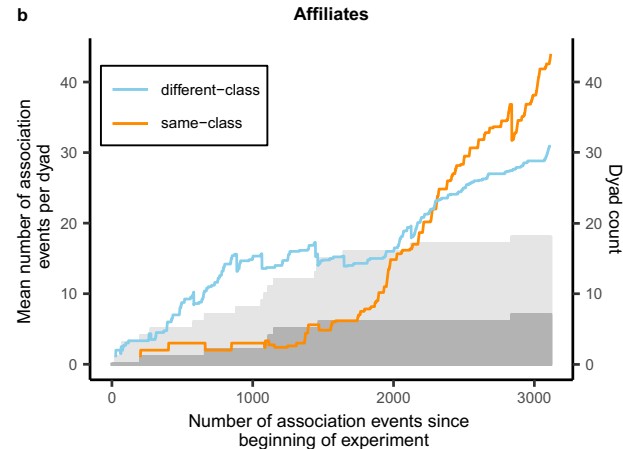

**Fig. 5 | Change in quantity and activity of different dyad types as the experiment progressed.** The mean number of events per dyad during each period of the experiment is displayed for same- (orange line) and different-class (blue line) **a** non-affiliate and **b** affiliate dyads. The number of same-class dyads (dark grey shading) is also shown alongside the combined number of same- and different-class dyads (light grey shading). Different-class affiliate dyads and both types of non-affiliate dyad were first observed shortly after the start of the experiment, but same-class affiliate dyads were only observed after approximately 200 association events had taken place. For affiliate dyads, the balance of events per dyad across same-class and different-class dyads shifted as the experiment progressed, indicating an increase in activity within same-class affiliate dyads during the last third of the experimental period. However, due to a greater number of different-class (11) as compared to same-class (7) affiliate dyads, this increase in within-dyad activity did not translate into an overall difference in likelihood of observation (308 observations of same-class affiliate dyads, 340 observations of different-class affiliate dyads). In contrast, same-class and different-class non-affiliate dyads displayed close to equal activity through the first approximately 10% of the experiment (300 events: 60 same-class non-affiliate dyads, 69 different-class non-affiliate dyads, 140 same-class observations, 144 different-class observations), but same-class activity exceeded different-class activity across the remainder of the experiment (2817 events: 333 same-class non-affiliate dyads, 339 different-class non-affiliate dyads, 1279 same-class observations, 906 different-class observations).

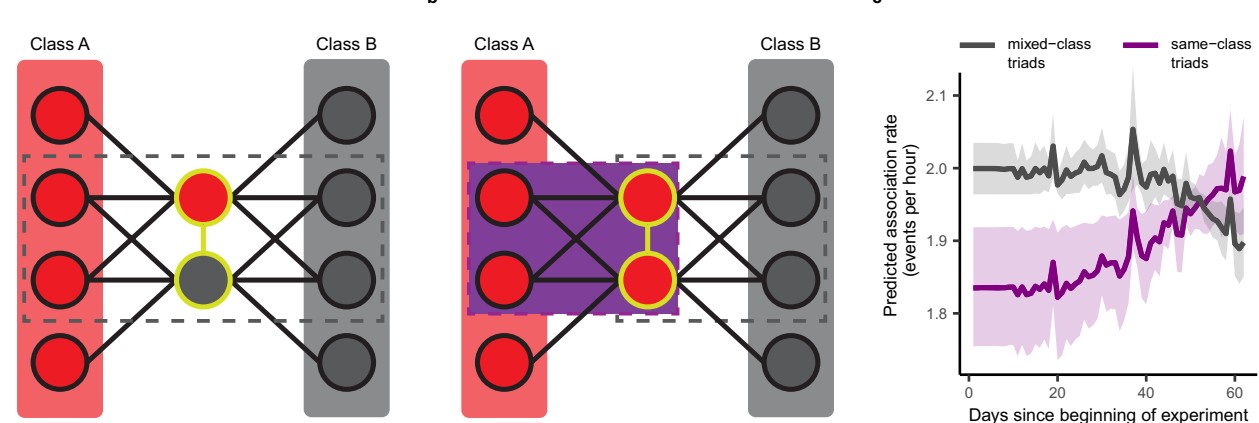

**Fig. 6 | Convergence in the social neighbourhoods of compatible task participants over the duration of the experiment.** When individuals retain ties with compatible group members and sever ties with those that are incompatible, then compatible individuals will increasingly share common social partners, resulting in formation of self-sustaining clusters of compatible group members within a social network[6–9]. **a** For a dyad (gold outline) in which the two members belonged to different treatment classes (A or B), selecting an individual from within the subset of all task participants with which both members of the dyad have associated yields a triad in which only two of the three individuals belong to the same class (grey dashed box). **b** When both members of a dyad belong to the same class, another subset can be defined which contains participants that have associated with both members of a dyad and belong to the same treatment class as them both (purple box). If associations occur according to a random process, same-class-only triads should be observed less frequently than mixed-composition triads (Supplementary Note 4: Expected triad composition in experimental networks and Supplementary Fig. 4). By summing over all triads in each subset, statistics can be calculated that indicate both a baseline likelihood of observing dyads that share common associates of any class, and the likelihood of observing same-class-only groupings (Supplementary Note 5: Triadic closure statistics in Eventnet 0.5.2). **c** The relationship between the class composition of a dyad's common associates and the dyad's predicted rate of future association changed as the experiment progressed. Same-class dyads for whom all three of their common associates belonged to the same class as both dyad members (purple) became more likely to be observed (though the magnitude of the effect was small). The opposite trend in predicted association rate was apparent for same-class dyads for whom none of their three common associates belonged to the same class as the two dyad members (grey). Median predicted incidence rate (solid line) and 95% prediction interval (shading) are displayed (for further details see Supplementary Methods: Visualization of REM output).

activity within these fully compatible groupings (as compared to groupings containing incompatible pairings) was only prevalent towards the end of the experiment (Fig. 6). Although clustering according to treatment class was therefore detectable long-term, the failure of affiliated individuals to adjust their associations to obtain higher rewards likely dampened the rate of progression of clustering sufficiently as to render it almost irrelevant as a signature of network re-structuring over short timescales. Even though we likely under-estimated the true number of affiliate dyads (as not all long-term relationships were known), the known affiliates still accounted for a disproportionately high number of observed interactions (20.8% [648 out of 3117] events despite comprising only 2.4% [18 out of 751] dyads; binomial test: $p < 0.001$). Given the prevalence of interactions involving members of affiliate dyads, their lack of individual plasticity will have substantially constrained the plasticity of the social network as a whole.

## Discussion

Our field-based results have important implications for a full under-standing of social cognition. First, whereas cognitive research often focuses on identifying specialized human-like traits (e.g., under-standing the causal role of a partner) that may underpin social and cooperative interactions in other animals[35], our findings imply that processes of individual discrimination and associative learning are sufficient to enable adaptive social plasticity[36,37]. We found no evidence that jackdaws understood the role of their partner in obtaining rewards (c.f.[35,38]), as improvement in coordination (i.e., synchroniza-tion of activity) between task partners was not linked to the experi-mental treatment. Nevertheless, given the dynamic nature of the task, with multiple birds often present near the apparatus at the same time (see Supplementary Note 2, Supplementary Movie 1 and Supplemen-tary Table 9), the ability of jackdaws to track and adjust their social associations with non-affiliates is likely to entail substantial information-processing demands[37]. Theoretical and empirical work shows that social assortment can arise through simple behavioural rules such as associating by phenotype[39] or leaving groups containing uncooperative members[19,40] but in our experiment such heuristics would not have increased individuals' rates of successful interaction. Instead, the mixed and changeable treatment-class composition of collections of foragers necessitated behavioural responses to the value of individuals. Specifically, to improve their performance jackdaws would need to have recognized multiple individuals within dynami-cally changing social environments, associated task partners' identities with reward outcomes, retained these learned associations and then used them to inform future partner-choice decisions. Consequently, the fidelity with which individuals can recognize group-mates and associate reward outcomes with individual identity will influence the potential for social plasticity[41]. Second, in line with other recent findings[42], our results suggest that under natural conditions indivi-duals are likely to forego potential short-term benefits to retain asso-ciations with valuable long-term partners. Theories of cognitive evolution often focus either on the challenges of selectively manip-ulating social interactions for short-term gain[2] or maintaining long-term fitness-enhancing relationships[43,44] but seldom address the trade-off between the two. Our experimental approach provides a tractable means to examine the changes in social-network topography linked to both factors, providing insights into how broad-scale social-network plasticity co-evolves with the partner-choice preferences of individuals.

Constraints on the plasticity of individual behaviour in turn have implications for how social conditions favourable to the persistence of cooperation between non-kin can arise and be sustained. While influ-ential theoretical models[6,7] have long argued that plastic social deci-sions generate assortativity within social networks, thus favouring cooperation, they ignore the existence of pre-existing relationships.

Our findings suggest such relationships can be crucial: the emergence of clusters of compatible jackdaw participants was detectable (likely as a by-product of feedback between individual learning and the social environment[9]), but it was not a prominent network characteristic. Thus, the magnitude of this effect is probably insufficient for it to be a key determinant of jackdaw social network structure, but its relevance for other social systems remains to be determined. Long-term social relationships are a feature of many animal societies, including our own, but the extent to which the fitness outcomes of partners is inter-dependent can vary substantially[45]. Jackdaw societies centre around long-term, genetically monogamous relationships, which accounted for 90% of recorded events between affiliates (580 out of 648 events). As mating partners have a strong stake in each other's fitness, they may be more constrained to associate together than in species with high levels of extra-pair mating or re-pairing[26,44,45]. In addition, continued association between juvenile siblings as well as between parents and their offspring is common in jackdaws during the months following the emergence of juveniles from the nest[27]. Consequently, the stability of jackdaws' social relationships likely limits the scope of network re-wiring via self-organization. Applying similar experimental techniques across a range of social systems is now necessary to determine the importance of this process as a force for promoting cooperation in nature.

When the value of social partners within a population varies, strategic adjustments of social associations may allow individuals to maximize their gains. Technological developments now provide the opportunity to examine plasticity in individual partner-choice and social network structure simultaneously in natural populations. By doing so, we find that processes of discrimination and individual learning, manifesting in the adjustment of social associations, enable jackdaws to exploit changes in their social environment. Importantly, our findings indicate that social network plasticity can be constrained by long-term relationships between group members with inter-dependent fitness. Recent empirical and theoretical work has high-lighted the value of viewing social structure as fluid, emerging from the feedback between the decisions of group members and the fine-scale social context in which they are made[46,47]. Our work provides impor-tant insights into the nature of this relationship in natural conditions and thus contributes to our understanding of the emergence of social environments conducive to the evolution of cognition and cooperation.

## Methods

### Study species and site

Jackdaws (*C. monedula*) are socially and sexually monogamous[26,48], colony-breeding corvids[27]. Breeding partners form long-term pair bonds, cooperate to build nests[49] and rear young, and coordinate to maintain close proximity during group movement[50]. Jackdaws use social information[29] and engage in both agonistic[51] and affiliative[52,53] behaviours during social foraging.

The experiment was conducted in 2019 at a breeding site in Sti-thians, West Cornwall, UK (N 50° 11′ 25.98″, W 5° 10′ 49.00″). During the breeding season, approximately 40 pairs rear young in nest-boxes located across the site. In addition, other pairs breed at the site but do not use the nest-boxes, and individuals that breed in nearby colonies also visit the areas in or around Stithians to forage. Over 90% of indi-viduals that occupied nest-boxes had been colour-ringed for individual identification before the start of the experiment, along with hundreds of other individuals that did not occupy nest-boxes. Nestlings reared in nest-boxes were ringed on the 25th day after hatching, which was approximately 10 days before fledging. Adults were caught for ringing using walk-in traps or remote-controlled traps fitted to nest-boxes, and ringing continued throughout the duration of the experiment. Each individual was fitted with three coloured leg rings, one of which con-tained a unique Radio-Frequency Identification (RFID) tag that could

be detected by an RFID data-logger. A metal leg ring displaying a British Trust for Ornithology (BTO) code was also fitted. Blood samples were taken during ringing for determination of sex[54].

Experimental work adhered to the guidelines of the Association for the Study of Animal Behaviour (ASAB) and the University of Exeter Biosciences Ethics committee (2014/577; eCORN000406). Ringing protocols were covered by Home Office (PPL 30/3261) and BTO (C6079, C5752, C5746) licences.

### Experimental design

**Treatment class assignment.** Each individual was assigned to one of two treatment classes (A or B) by either supervised or unsupervised randomization. The supervised randomization procedure was used to assign treatment class at random whilst constraining the assignment to ensure an equal number of individuals in each class. Supervised randomization was applied separately to two different categories of dyad or individual prior to commencement of the experiment. First, known breeding pairs in which both individuals possessed a working RFID-tag were assigned such that there were equal numbers of these pairs in the same treatment class (AA/BB) or in different treatment classes (AB). Then, regular feeder users (as determined from inspection of passive RFID feeder data prior to the breeding season) that did not have a known partner were assigned such that an equal number of regular users belonged to each class. All remaining individuals that had been fitted with an RFID-tag prior to the commencement of the experiment were assigned to treatment classes using an unsupervised randomization procedure, meaning that the total number of these individuals assigned to each class was not controlled. Jackdaws that were ringed during the experiment, including fledging juveniles, were also assigned to treatment classes in this fashion, such that the numbers of juveniles within each treatment class, both within a specific brood and in the population as a whole, was randomly determined. As a result, interactions between siblings (for this experiment, defined as newly fledged juveniles from the same brood), and between parents with their offspring, could feature any combination of classes.

**Dual-feeder task.** We developed an automated dual-feeder task to assess the flexibility of partner-choice decisions by wild jackdaws. Two identical task apparatuses were placed within the site, separated by approximately 100 m. The task apparatus (Fig. 1a) comprised two feeder units, with the feeding platforms separated by approximately 0.5 m. Within each feeder unit, there was a low-quality and a high-quality reward. The results of a food preference test verified the relative quality of the two rewards (see Supplementary Note 1). The low-quality reward was grain, whereas the high-quality reward was a 50/50 mix of grain and crushed, dried mealworms. When individuals visited the feeder units, they occupied a perch that contained an RFID antenna which was connected to an IBT EM4102 data logger. This RFID reader recorded the unique RFID-tag code of each visitor, each second during which the visitor occupied the perch, and which perch the individual occupied. In turn, the RFID reader was connected to a 'Darwin Board' microcomputer, which processed the RFID-tag data to determine whether the state of the task should change. When a change in task state occurred, the 'Darwin Board' operated servo motors fitted within feeder doors located adjacent to the perch, so altering the accessibility of the rewards.

Foragers could access the low-quality reward if foraging alone. Including this reward for solo foraging activity was necessary to encourage feeder exploration, so as to overcome initial neophobia[29] towards the feeders. When foraging simultaneously with another member of the same treatment class, both individuals received access to the high-quality reward that was inaccessible during solo feeder usage. Access to the high-quality reward was provided for a minimum of 15 s from the onset of simultaneous feeder occupation by a same-class dyad. If a successful dyad continued to occupy their feeding

positions for longer than 15 s, then the high-quality reward remained accessible to both participants until one of the individuals left or was replaced by another forager. The onset of simultaneous foraging by individuals that were assigned to different treatment classes triggered a period of feeder inactivity. During this 'lockout' period, neither type of reward was accessible unless simultaneous foraging by a same-class dyad occurred, in which case the high-quality reward was accessible, as described above, until the end of the event, at which point the feeder returned to the lockout state. Each lockout lasted a minimum of two minutes; further unsuccessful dyadic foraging events during the lockout period would trigger a restart of the lockout. At the end of a lockout period, the feeder returned to its default state (low-quality reward accessible, high-quality reward inaccessible). In total, there were four different task states and eight possible state transitions (Fig. 1b).

### Data collection

The data collection period was from the beginning of April until the end of July 2019. The experiment was run on five days per week, beginning at approximately 06:00 and ending at approximately 10:00 each day. At the beginning of each experimental period, the dual-feeder tasks were tested using spare RFID-tags (with treatment class assignments) to ensure task functionality. In addition, during each period the task was checked hourly to ensure that the food rewards remained stocked. RFID 'test-tag' records were subsequently removed from the dataset during initial data processing (Supplementary Methods: RFID data processing). Other sources of food that we provided at the site (i.e., other bird feeders) were not accessible during the periods in which data collection from the task occurred. Outside of the experimental windows, the task was covered such that birds could not interact with it. Prior to statistical analysis, the data were processed to improve estimates of event duration (Supplementary Methods: RFID data processing).

### Statistical analysis

**Relational Event Models.** Relational Event Models (REMs) are a form of time-to-event analysis that incorporates social interaction, where a 'relational event' is a dyadic interaction directed from one entity ('source') to another ('target') that occurs at a specific point in time[31,32]. REMs do not require any aggregation of data prior to analysis, making them particularly useful for studying fine-scale change in social behaviour over time. In addition, REM analysis can be used to investigate hypotheses relating to multiple levels of social structure (individuals, dyads, network-level properties) and relationships therein. For social association events, we defined 'source' and 'target' by time of arrival at the feeder, with the 'source' being the first individual of the dyad to arrive and the 'target' being the individual that subsequently joined. We performed the REM analysis as a two-step process. First, the time-varying covariates (details in Supplementary Methods: REM specifications, see also Supplementary Table 10 for an explanation of general and model-specific REM terminology) required for the REM were generated using Eventnet (version 0.5.2)[55]. These covariates included 'triadic closure' statistics, which represented the rate of association between individuals that shared common task partners (for details see Supplementary Note 5). Then, after dataset preparation, REMs were fitted by applying Cox proportional hazards models to the processed dataset using the 'coxph' function from the 'survival' package (version 2.44)[56] in R[57]. For each REM, a 'strata' term was included to allow the baseline probability of observing an association event (i.e., the baseline hazard) to vary depending on the day in question. We fitted separate REMs where it was necessary to investigate hypotheses that either required null models generated by different permutation procedures, in cases where similar hypotheses were investigated at different levels of analysis (e.g., individual versus dyad level), or if the use of multiple models was required to preserve model interpretability. All results are

reported in terms of relative risk, which was obtained by calculating hazard ratios (hereafter referred to as 'Incidence Rate Ratios': IRR) from model coefficients (by application of the exponential function to REM linear predictors).

**Permutation procedures.** The structure of social network data and the sampling biases often associated with collecting it can invalidate the use of common forms of statistical analysis, so permutation tests are typically used for hypothesis testing[58]. In practice, permutation tests applied to social network data are used to determine the significance of a model term of interest by comparing a value of this term from a model fitted to unpermuted data to a distribution of values of that term extracted from models fitted to permuted data. Conceptually, a typical REM analysis works by comparing each observed relational event to a set of associated 'non-events', where these non-events are generated by drawing 'source' and 'target' labels at random from the total label-set of the dataset, with an equal probability of each label being drawn (i.e., a uniform probability distribution across labels). However, this type of procedure was not appropriate given the nature of our experiment. For example, juveniles that fledged during the experiment were only active participants in the latter weeks, so constraints needed to be applied to prevent the generation of permuted datasets featuring the presence of juveniles in the early stages of the experiment. Consequently, we adapted a data-stream permutation procedure[59–61] for use with REMs to determine the significance of model coefficients. By utilizing data-stream permutation in conjunction with REMs, we tested whether the performance of individuals or dyads exceeded that expected had the treatment-class combinations of participants occurred at random. The null hypothesis was therefore that patterns of treatment class combination, rather than social interactions per se, could be adequately described by a random process. Because each individual's treatment class was randomly assigned, a continuation of pre-experiment association patterns, and therefore no response to the experimental treatment, would give rise to random treatment class combinations.

Permutation of source and target labels was achieved using a combination of shuffling of 'source' and 'target' labels and constrained label swaps. We chose to primarily employ a 'shuffle' randomization (i.e., randomized re-ordering of the labels within a set) to minimize the use of swaps (i.e., switching the labels of a pair of entries at random), which can produce sub-optimal randomization when constraints on allowable swaps are applied (see Supplementary Note 6). Randomization was applied independently to subsets of the data defined by each combination of sampling period and location (i.e., day and feeder). Unconstrained shuffling of source and target labels generated instances in which the same individual was labelled as both the source and the target for a given record (a 'loop'). If such loops were detected in a subset of the permuted data following shuffling, the randomization was repeated until no loops were present or the maximum number of randomizations (10,000) had been performed (Supplementary Fig. 2). This randomization limit was selected following permutation procedure validation (see Supplementary Methods: Permutation procedure validation). If the randomization limit was reached without producing an acceptable collection of labels, constrained pairwise swaps were then performed on the most recent permuted subset to remove loops. For each loop in turn, another instance was selected at random from the permuted subset. If the chosen event did not feature the individual in the loop as either the source or target, then either the source or target labels (chosen at random) were swapped between the loop instance and the other, non-loop instance to create two new valid, non-loop instances (Supplementary Fig. 2).

There are four variants of the above procedure that can be used with REM datasets to produce different types of null model for hypothesis testing purposes (Supplementary Fig. 3). These variants differ in the aspects of network structure that they preserve, such that

the appropriate variant depends on the hypothesis being tested. For example, employing non-independent randomization of source and target labels (Supplementary Fig. 3a) was appropriate for testing whether the likelihood of successful association between dyads significantly exceeded that expected from a model of random treatment class combination, but would be inappropriate for testing whether individuals that joined association events ('targets') improved their task performance at a greater rate than individuals that initiated such events ('sources'). In this scenario, independent randomization of source and target labels (Supplementary Fig. 3b) is more appropriate, as each individual's number of events as source and target is preserved. Therefore, to examine the factors influencing individual performance on the task, independent randomization of source and target labels was used (for Models 1 & 2), but to test whether patterns of dyadic interaction and network structure significantly differed from that expected given random combination of participants' treatment classes, we used non-independent randomization of source and target labels (Supplementary Fig. 3a) to generate permuted datasets (for Models 3, 4, 5 & 8; for details of each of the models used, see Supplementary Methods: REM specifications). Finally, to examine coordination of association event duration and arrival times (Models 6 & 7), permutation of edge weights was performed (Supplementary Fig. 3d). This procedure did not feature shuffling or swapping of source or target labels, nor alteration of edge direction. Instead, only the edge weights were permuted. This randomization was performed by shuffling edge weights (event duration or arrival latency) between events, within subsets defined by day and feeder, using the sample function in R and did not require the use of swaps. Ten thousand permuted datasets were generated using each of the three permutation procedure variants described above.

**Permutation tests.** To assess whether the observed proportion of successful association events exceeded that expected should treatment-class combinations have occurred at random, the proportion of successful events in the unpermuted dataset was compared to a 95% confidence interval of equivalent proportions calculated from 10,000 permuted datasets (generated by non-independent randomization of source and target labels). This confidence interval was derived from calculation of the 2.5% and 97.5% quantiles of the distribution of proportions from permuted datasets. Similarly, the day-to-day change in the difference between overall numbers of successful and unsuccessful association events was calculated for unpermuted and permuted data to illustrate how the observed event type differential departed from that expected (should treatment-class combinations occur at random) during different stages of the experiment (see Fig. 3). The median expected event type differential and a 95% confidence interval (as above) were calculated from the permuted data.

**Permuted Relational Event Models (pREMs).** Once the collections of permuted datasets had been generated, each of the datasets was spliced with the original, unpermuted data, such that hybrid datasets were produced comprising the observed events (unpermuted) and the 'non-events' generated by the appropriate permutation procedure. These datasets were structured such that each event was followed by its lone, accompanying 'non-event', forming an alternating series of events and 'non-events'. Following the addition of the time-varying covariates computed in Eventnet, REMs were fitted using each of these hybrid datasets. In a REM analysis applied to animal behaviour data, properties of the individuals in each of the observed events are compared to the corresponding 'non-events', such that the resultant model coefficients represent the extent to which the observed events (unpermuted data) differ from the set of expected events given the sampling specifications (permuted data). For each fitted model, the coefficient value for each term was extracted and stored. Combining the coefficient estimates from all models then produced a distribution

of coefficient values for each term. Quantiles of each coefficient distribution were then used to obtain coefficient estimates and confidence intervals. The median coefficient value was taken as the value most representative of the magnitude and direction of the effect of a REM term. The statistical significance of each REM term was determined by inspection of zero-crossing of the raw REM coefficients; if the confidence interval of the term spanned zero, then the effect was deemed non-significant. For REM analyses not concerning individual-level hypotheses (i.e., analyses that addressed dyad-level [including coordination] or network-level hypotheses), 95% confidence intervals were calculated from quantiles of coefficient distributions (i.e., 2.5% and 97.5% quantiles). The REMs that addressed individual-level hypotheses used two separate models to test similar hypotheses using the same datasets. Therefore, to account for multiple testing, the confidence intervals were widened for these two individual-level models from 95% to 97.5% (i.e., 1.25% and 98.75% quantiles) and statistical significance was assessed by checking for boundary-crossing as above. Widening the confidence intervals by this extent provided a correction for multiple testing akin to halving the p-value threshold (i.e., Bonferroni correction). Finally, predicted values were extracted from each model for the purpose of visualizing REM output (see Supplementary Methods: Visualization of REM output).

### Reporting summary

Further information on research design is available in the Nature Portfolio Reporting Summary linked to this article.

## Data availability

The data that support the findings of this study can be accessed from Figshare (https://figshare.com/collections/Cornish-Jackdaws/6723399; https://doi.org/10.6084/m9.figshare.c.6723399). We provide the raw.csv files downloaded from the task apparatus after each session, the processed data combined into a single.csv file, the permuted REM datasets, the datasets required to reproduce the figures, and documentation on the uses of each dataset type and their place in the data analysis workflow. The Figshare repository also contains video clips illustrating the dynamic nature of social activity during engagement with the dual-feeder task.

## Code availability

Eventnet 0.5.2 and Eventnet tutorials are available from Github (https://github.com/juergenlerner/eventnet). The C code for 'Darwin Board' microcomputer programming and R code used for data processing and analysis are also available from the Github repository for this study (https://github.com/mkings-220920/Cornish-Jackdaws), https://doi.org/10.5281/zenodo.8105897

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

## Acknowledgements
Special thanks to Odette Eddy, the Gluyas family and everyone at Pencoose Farm. Cindy Maslarov, Jennifer Austin and Emma Doyle assisted with preliminary tests of the experimental apparatus. Victoria Lee and Alison Greggor assisted with the ringing of the jackdaws. Matt Silk and Hilary Reed provided comments that improved the manuscript. M.K. and J.J.A. were supported by BBSRC South West Doctoral Partnership (SWDTP) studentships (codes 630051486 and 680027356 respectively). J.L. was supported by Deutsche Forschungsgemeinschaft (DFG) – 321869138. A.T. and G.M. are supported by a Leverhulme Trust grant (RGP-2020-170) to A.T.

## Author contributions
M.K., A.T. and A.R. conceived and designed the experiment and A.T. supervised the work; M.W. designed the 'Darwin Board' microcomputer; M.K. and M.W. designed the feeder apparatus and programmed the 'Darwin Board' microcomputers; M.K., J.J.A., G.M., and M.W. constructed the feeder apparatus; G.M. maintained field sites/study populations and ringed the jackdaws with help from J.J.A.; J.J.A. collected the data, provided contributions that improved the task design and statistical analysis, and produced illustrations; M.K. and J.L. designed the statistical analysis; M.K. performed the statistical analysis and drafted the manuscript. All authors discussed results, contributed critically to drafts and gave final approval for publication.

## Competing interests
The authors declare no competing interests.
