## [Peer Review File · Nature Communications]

REVIEWER COMMENTS

Reviewer #1 (Remarks to the Author):

The current manuscript examines the ability for jackdaws to adjust their patterns of social association based their success rate in interacting with an ingenious automated social foraging apparatus in the wild. The authors report that the birds alter their pattern of association, favouring collaboration with compatible social fellows and disfavouring association with incompatible birds based on their success interacting with the foraging task. These changes in association preferences do have measurable, albeit subtle, effects on the overall structure of the social network. The effect of these social adjustments is relatively small because the jackdaws do not alter their tendency to interact with individuals that they share a strong pre-existing social bond with, for example breeding pairs. This suggests that long term social relationships take precedence over local optimisation of foraging success in this species. The results suggest to me that animals may be able to optimise social interactions (to some degree) through the relatively simple mechanisms of individual recognition and associative learning without more complicated social memory/accounting being necessary. I think it would be interesting to do something similar in a species with relatively weaker enduring social bonds as pair bonding and familial bonds clearly take centre stage with the jackdaws.

I think this is an innovative approach to studying social network plasticity as it leads to alterations in social network structure in wild animals without the relatively coarse manipulation of adding or removing individuals. I think the results are compelling and will be of broad appeal to those interested in the social dynamics of group living animals. The paper is well written and clear, I have few suggestions for improvement. I do think that adding legends to the figures would help the reader interpret them more easily rather than having to refer to the captions.

I am not deeply familiar with the statistical approach taken here, though it is well explained, and does seem to be sensible and well supported.

Reviewer #2 (Remarks to the Author):

I enjoyed reading this Ms in which the authors describe a field experiment with jackdaws in which they manipulate the association patterns of individuals based on their experience at a feeder. The found that association patterns of jackdaws could indeed be changed but only to a small extent showing both the ability to "rewire" networks as well as the importance of long-term associations. This is a thought-provoking experiment and I am not sure I understood all the evidence correctly. Does the rewiring of associations purely come from the time the birds spent at these feeders or is there evidence that they also spent time with "compatible" conspecifics in other places and contexts? If yes, this would be quite interesting because it would show that the birds generalize from their experience at the feeder. If not, it would appear to be some sort of Skinner-box operant conditioning that the jackdaws merely learn to turn up at the feeder when a particular bird is present.

In other words it would be very important to know where and when these compatible birds were together. It was not clear to me from the way the data were presented whether the authors can even make this sort of distinction. If all the association data were taken at or near the feeders then the authors probably cannot make this type of distinction. This has been a weakness of similar studies on bird table behaviour by the Sheldon group in Oxford where the ecological relevance of the association patterns of birds away from the bird table is often unknown (but obviously very important).

I was also wondering about the design of the experiment. The birds don't actually need to interact with the compatible individual(s) in order to gain the benefit. Therefore it is difficult to see how an association preference should actually be learnt. Maybe it is a byproduct of other preferences such as the fact that a compatible bird likes to visit the feeder at a particular time of day and the matching bird then learns to feed during that time rather than with this bird.

The references are largely focused on association patterns in birds and there are some on mammals. Surprisingly, none of the papers on re-wiring in fish (guppy) social networks were cited. Darren Croft and his team have published extensively on this topic in the context of cooperation.

Reviewer #1 (Remarks to the Author):

The current manuscript examines the ability for jackdaws to adjust their patterns of social association based their success rate in interacting with an ingenious automated social foraging apparatus in the wild. The authors report that the birds alter their pattern of association, favouring collaboration with compatible social fellows and disfavouring association with incompatible birds based on their success interacting with the foraging task. These changes in association preferences do have measurable, albeit subtle, effects on the overall structure of the social network. The effect of these social adjustments is relatively small because the jackdaws do not alter their tendency to interact with individuals that they share a strong pre-existing social bond with, for example breeding pairs. This suggests that long term social relationships take precedence over local optimisation of foraging success in this species. The results suggest to me that animals may be able to optimise social interactions (to some degree) through the relatively simple mechanisms of individual recognition and associative learning without more complicated social memory/accounting being necessary. I think it would be interesting to do something similar in a species with relatively weaker enduring social bonds as pair bonding and familial bonds clearly take centre stage with the jackdaws.

Thank you for this positive appraisal and summary of our findings. We agree that it would be interesting to examine social plasticity in species with different social systems, and we highlight the importance of such research and how our experimental approach can facilitate it (lines 211-236).

We are glad to see the reviewer noting the importance of “relatively simple mechanisms” because, as we point out on lines 190-191, cognitive research often has an anthropocentric bias to search for human-like skills. An important point to note here is that, given the dynamic nature of social interactions, the information-processing demands of adjusting social associations strategically (even via ‘simple mechanisms’ such as individual recognition and learning) are substantial (lines 196-209). We have also added a new sentence highlighting how the fidelity of individual recognition influences the potential for social plasticity (line 207).

I think this is an innovative approach to studying social network plasticity as it leads to alterations in social network structure in wild animals without the relatively coarse manipulation of adding or removing individuals. I think the results are compelling and will be of broad appeal to those interested in the social dynamics of group living animals. The paper is well written and clear, I have few suggestions for improvement. I do think that adding legends to the figures would help the reader interpret them more easily rather than having to refer to the captions.

We are delighted that the reviewer found our approach innovative and the results compelling. We have added labels and/or legends to Figures 1, 4, 6, 7 & 8 to aid interpretability. In addition, the colour schemes of Figures 8 & 9 have been changed to avoid the use of red and green colour combinations.

I am not deeply familiar with the statistical approach taken here, though it is well explained, and does seem to be sensible and well supported.

We are glad that the reviewer found the approach to be well explained and justified. To clarify the conceptual basis of the modelling approach, we have added a section further detailing the null hypotheses tested by the relational-event models that we used (lines 375-382).

Reviewer #2 (Remarks to the Author):

I enjoyed reading this Ms in which the authors describe a field experiment with jackdaws in which they manipulate the association patterns of individuals based on their experience at a feeder. The found that association patterns of jackdaws could indeed be changed but only to a small extent showing both the ability to “rewire” networks as well as the importance of long-term associations.

Thank you for this positive appraisal and summary.

This is a thought-provoking experiment and I am not sure I understood all the evidence correctly. Does the rewiring of associations purely come from the time the birds spent at these feeders or is there evidence that they also spent time with “compatible” conspecifics in other places and contexts? If yes, this would be quite interesting because it would show that the birds generalize from their experience at the feeder.

The question of whether the changes in association we see at experimental feeders generalise to other contexts is interesting, but this was not the focus of our experiment and is not relevant to the interpretation of our results. Importantly, there is no *a priori* reason to expect these manipulations to extend to other contexts. The experiment clearly demonstrates that jackdaws can adjust their social foraging associations flexibly based on payoffs. Given this plasticity, it is not clear why responses to foraging payoffs should carry over into other contexts, where payoffs will likely be different. Indeed, theory suggests that reputational benefits across contexts should only be expected to arise where there is limited plasticity. We highlight these issues and the need for further research on lines 124-129.

Our goal was not to investigate cross-contextual reputation but to determine whether individuals in a natural, freely interacting society can learn to adjust their associations to maximise rewards (lines 57-59), which has never before been addressed (lines 17-19, 54-57). By providing the first evidence that individuals adjust social associations according to payoffs, we address a previously untested assumption of influential theories of cognitive and social evolution (lines 211-213, 219-221). Thus, our paper presents findings that are an important advance in their own right (lines 240-248) as well as detailing a tractable experimental approach that can be used in future work (lines 234-236).

We also aimed to determine the network-level consequences of social plasticity (lines 163-187). This too is novel because no previous study has quantified the magnitude and trajectory of network plasticity in response to changing social payoffs in a natural population (lines 45-57). Moreover, our results are of clear theoretical importance. The theoretical literature (e.g., Evolutionary Graph Theory) has long emphasised potential links between payoff structures, network dynamics and the persistence of cooperation in social groups, but has overlooked the fact that individuals may have stable, pre-existing relationships. By revealing that the need to maintain long-term relationships constrains network plasticity, we highlight an important

missing component of the theoretical literature, which will help spur future empirical work on other species. We now make this point on lines 219-222.

In conclusion, we present evidence (i) that individuals in natural, freely interacting societies learn to adjust their associations to maximise rewards (lines 106-124) and (ii) that pre-existing social relationships affect social plasticity (lines 130-152), as well as (iii) quantifying the effects of individual partner-choice decisions on network structure for the first time in a wild population (lines 163-187). All these findings are novel and do not require knowledge about whether social changes generalise to other contexts – that is indeed an interesting question to tackle, but a separate one for future studies.

...If not, it would appear to be some sort of Skinner-box operant conditioning that the jackdaws merely learn to turn up at the feeder when a particular bird is present.

“Skinner box” operant conditioning would be insufficient to explain our findings. In stark contrast to a Skinner box task, our task was extremely dynamic: individuals did not just visit at specific times, but rather throughout the periods when the task was in operation, and patterns of social foraging reflected the fission–fusion structure of jackdaw societies. For example, trials ran on average for 4 hours each morning (mostly between 0600 and 1000). If we split the full range of times that trials were run across (~0500-1200) into half-hour windows (e.g., 0500–0530, 0530–0600) then, on average, each individual visited the task across a mean of 8.0 distinct time windows. Similarly, for each unique 5-minute window in which there was a social association, we detected a mean of 3.6 unique birds, with a maximum of 14. This highlights the dynamic and variable nature of task visitation. Thus, individuals had to keep track of numerous potential participants and their respective payoff values simultaneously within a changeable social landscape. We have edited the text to clarify these points (lines 102-104, 196-197 and 205), and we provide additional data on patterns of visitation (Supplementary Material: Descriptive Statistics) and an annotated video highlighting the dynamic nature of the task.

In other words it would be very important to know where and when these compatible birds were together. It was not clear to me from the way the data were presented whether the authors can even make this sort of distinction. If all the association data were taken at or near the feeders then the authors probably cannot make this type of distinction. This has been a weakness of similar studies on bird table behaviour by the Sheldon group in Oxford where the ecological relevance of the association patterns of birds away from the bird table is often unknown (but obviously very important).

As we note above, whether or not learned social associations generalise to other contexts is not pertinent to the core questions that we are tackling in this study. The data that we present provide clear evidence that jackdaws learn to adjust their associations based on the social value of partners in a particular context. Such associations would not necessarily be expected to carry over to different contexts, where payoffs will be different. Testing this possibility will be an interesting focus for future studies (lines 124-129).

I was also wondering about the design of the experiment. The birds don't actually need to interact with the compatible individual(s) in order to gain the benefit. Therefore it is difficult to see how an association preference should actually be learnt. Maybe it is a byproduct of other preferences such as the fact that a compatible bird likes to visit the feeder at a particular time of day and the matching bird then learns to feed during that time rather than with this bird.

Simple preferences or 'social heuristics' (i.e., rules of thumb), such as leaving groups containing uncooperative individuals, have been shown in lab experiments on guppies to benefit co-operators, by facilitating clustering of cooperative individuals within social networks. We now refer to this work (lines 199-201). However, in complex social systems where individuals may encounter a temporally changing range of potential partners and the payoffs of associating with particular partners vary, recognizing individual partners and updating appraisals of the value of social relationships can be beneficial. We now highlight this on lines 28-33. Our experiment was explicitly designed such that an individual's payoff depended on the identity of the specific partner that it engaged with. Therefore, learning occurs because jackdaws associate individual identities with reward outcomes: that is, the presence of a particular individual at the task is associated with rewards (or a lack of reward). We explain this on lines 202-207.

A social heuristic (as suggested by the reviewer), such as leaving the task when unfavourable outcomes were encountered, and joining other foragers at another time, could conceivably lead to improvement in individual performance at this type of task. But only if 1) jackdaws foraged in groups of consistent membership 2) the treatment-class composition of foraging groups was biased and 3) foragers/foraging groups engaged with the task at set times. If the above conditions are met, then an individual would benefit from leaving the task when outcomes are unfavourable and re-engaging with the task at different times until encountering a group in which its treatment class was shared with a significant majority of the group members. Then, associating indiscriminately with members of this group would result in the individual achieving net success at the task.

However, these conditions were not met. First, though dyadic association events often occurred as part of social foraging when groups of jackdaws were in the vicinity of the task, these groups were not cohesive and membership turnover was high. We now include supplementary data showing that the number of jackdaws near the task at any given time was highly variable and a supplementary video to highlight that foraging sessions typically featured individuals frequently arriving and leaving the vicinity of the task, rather than stable groups. Second, birds were randomly assigned to treatment classes. When coupled with a high rate of membership turnover, randomized treatment class assignment minimized the scope for bias in the treatment-class composition of groups to influence an individual's probability of success. Finally, as described above, individuals visited the task throughout the period of the day that it was active. Consequently, the use of a simple social heuristic, such as "approach feeder if another bird is present", "attend feeder at time x" or "leave group until rewarded", would not have resulted in an individual performing better than chance. We have added a section in the Discussion to explain this (lines 199-207).

The references are largely focused on association patterns in birds and there are some on

mammals. Surprisingly, none of the papers on re-wiring in fish (guppy) social networks were cited. Darren Croft and his team have published extensively on this topic in the context of cooperation.

We have now cited guppy papers from Darren Croft's lab on lines 50 and 199-201. These studies are certainly relevant, but it is important to note that they concern assortment by cooperative phenotype ('cooperativeness'). By contrast, our study addresses assortment resulting from selective adjustment of social ties among individuals in response to changing payoffs associated with dyadic social interaction. Unlike simple phenotypic assortment, such selective adjustment requires individual recognition and learning. We note the role of assortment by cooperativeness as a means by which assortativity can be generated in networks (lines 199-200), but explain that it could not have produced the assortment patterns detected in our experiment, and further emphasise that our experimental approach is designed to quantify the importance of learning (lines 56-57, 205-206, 240-242) and individual recognition (lines 204-209) in social network re-wiring.

REVIEWERS' COMMENTS

Reviewer #1 (Remarks to the Author):

I think the authors have done a good job responding to the reviewers and I have no further comments.

Reviewer #2 (Remarks to the Author):

The authors have carried out a thorough revision. Despite the fact that I was initially quite critical of some aspects of the paper, I feel that the authors have addressed all my queries in sufficient detail. Therefore I recommend acceptance of this Ms.